# Detecting AutoAttack Perturbations in the Frequency Domain

Peter Lorenz [1 2]  Paula Harder [1 3]  Dominik Straßel [1]  Margret Keuper [4]  Janis Keuper [5 1]

## Abstract

Recently, adversarial attacks on image classification networks by the *AutoAttack* (Croce & Hein, 2020b) framework have drawn a lot of attention. While *AutoAttack* has shown a very high attack success rate, most defense approaches are focusing on network hardening and robustness enhancements, like adversarial training. This way, the currently best-reported method can withstand $\sim 66\%$ of adversarial examples on CIFAR10.

In this paper, we investigate the spatial and frequency domain properties of *AutoAttack* and propose an alternative defense. Instead of hardening a network, we detect adversarial attacks during inference, rejecting manipulated inputs. Based on a rather simple and fast analysis in the frequency domain, we introduce two different detection algorithms. First, a black box detector which only operates on the input images and achieves a detection accuracy of $100\%$ on the *AutoAttack* CIFAR10 benchmark and $99.3\%$ on ImageNet, for $\epsilon = 8/255$ in both cases. Second, a white-box detector using an analysis of CNN feature-maps, leading to a detection rate of also $100\%$ and $98.7\%$ on the same benchmarks.

## 1. Introduction

The vulnerability of neural networks towards adversarial attacks is one of the key obstacles which are currently limiting the applicability of deep learning models for a wide range of practical use cases. While the latest attack methods, like *AutoAttack* (Croce & Hein, 2020a), are achieving very high success rates perturbing input data on Convolutional Neural Networks (CNN) based image classifiers, the avail-

[1]HPC, Fraunhofer ITWM, Germany [2]IWR, Heidelberg University, Germany [3]Scientific Computing, University of Kaiserslautern, Germany [4]Data and Web Science Group, University of Mannheim, Germany [5]Institute for Machine Learning and Analytics - IMLA, Offenburg University, Germany. Correspondence to: Peter Lorenz <peter.lorenz@itwm.fhg.de>.

*Accepted by the ICML 2021 workshop on A Blessing in Disguise: The Prospects and Perils of Adversarial Machine Learning.* Copyright 2021 by the author(s).

**Table 1** Results of the proposed detectors on *AutoAttack (standard mode)* for different choices of the hyper-parameter $\varepsilon$ (default in most publications is $\varepsilon = 8/255$) and test sets. *ASR=Attack Success Rate*, *ASRD=Attack Success Rate under Detection*. White-Box results on ImageNet are obtained by a Logistic Regression classifier, Random Forests were used in all other cases. Table 5 in the appendix is showing the full results for both classifiers. F1 and the False Negative Rate (FNR) are used to report the detection performance. See section 3 for details of the experimental setup.

| $\varepsilon$ | ASR | Black-Box | | | White-Box | | |
|---|---|---|---|---|---|---|---|
| | | F1 | FNR | ASRD | F1 | FNR | ASRD |
| CIFAR10 on WideResNet28-10 | | | | | | | |
| **8/255** | **100** | **98.2** | **00.0** | **00.0** | **99.0** | **00.0** | **00.0** |
| 4/255 | 100 | 93.8 | 00.3 | 00.3 | 96.4 | 05.0 | 05.0 |
| 2/255 | 93.1 | 85.0 | 05.3 | 04.9 | 85.8 | 05.0 | 04.6 |
| 1/255 | 49.6 | 70.8 | 22.7 | 11.3 | 62.5 | 37.3 | 18.5 |
| 0.5/255 | 12.3 | 54.8 | 46.7 | 05.8 | 61.6 | 52.0 | 06.4 |
| ImageNet on WideResNet51-2 | | | | | | | |
| **8/255** | **100** | **87.1** | **00.7** | **00.7** | **96.7** | **01.3** | **01.3** |
| 4/255 | 100 | 77.4 | 08.7 | 08.7 | 94.1 | 02.7 | 02.7 |
| 2/255 | 100 | 60.2 | 28.0 | 28.0 | 82.3 | 16.7 | 16.7 |
| 1/255 | 99.9 | 53.4 | 43.7 | 43.6 | 67.9 | 30.7 | 30.6 |
| 0.5/255 | 96.9 | 54.4 | 42.0 | 40.7 | 59.0 | 41.0 | 38.1 |

able defense methods appear to be "always one step behind". In general, adversarial defense strategies can be grouped into roughly two different approaches: First, the hardening of networks, which is mostly done via adversarial training, and second, the detection of adversarial samples during inference. In this work, we propose a simple feature-driven approach to detect adversarial examples, where features are extracted in the frequency domain, based on prior work by (Harder et al., 2021), which shows almost perfect detection results on state-of-the-art (SOTA) adversarial benchmarks.

### 1.1. Related Work

**The AutoAttack Benchmark.** In 2020, (Croce et al., 2020) launched a benchmark website[1] with the goal to provide a standardized benchmark for adversarial robustness. Until then, single related libraries such as FoolBox (Rauber et al.,

---

[1]robustbench.github.io

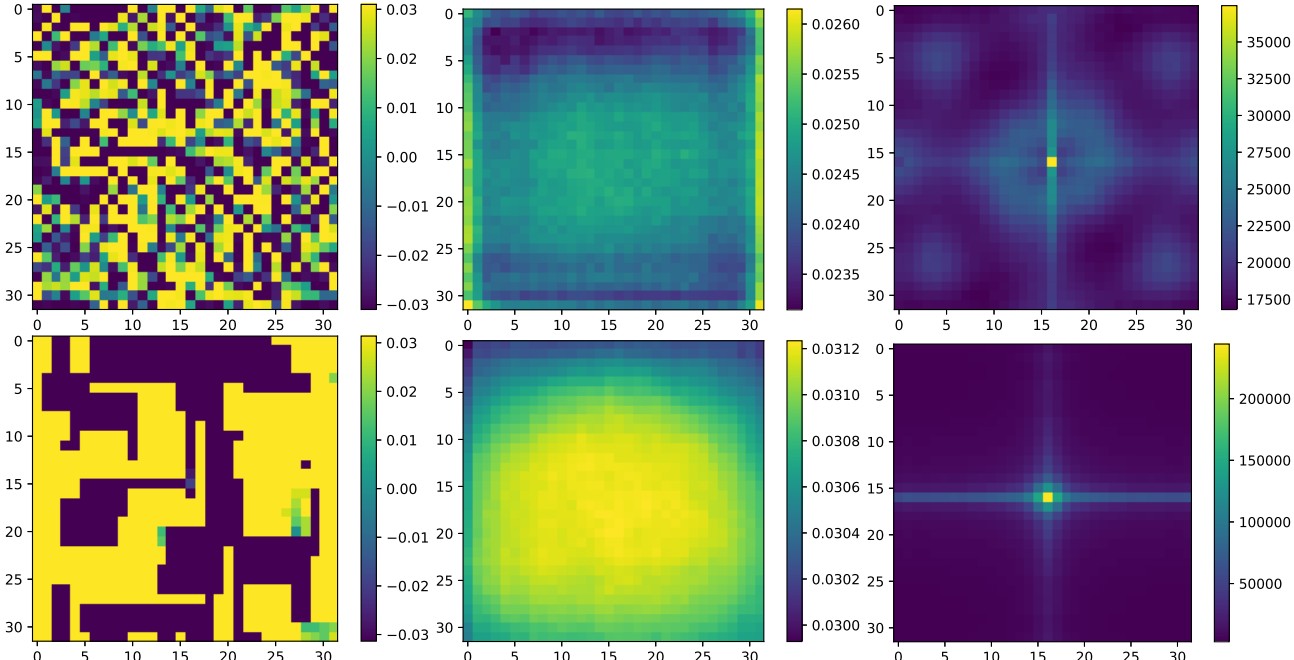

*Figure 1.* Visualization of AutoAttack perturbations on a ResNet18 for CIFAR10. The top row: APGD-CE $\ell_\infty$ attack, bottom row: Squares $\ell_\infty$ attack. Left column shows the spacial difference between a random test image from CIFAR10 and its perturbation. The center column depicts the mean of spacial differences over 1000 perturbed images. Right column: accumulated magnitudes of the spectral differences over the same 1000 images. While there are no obvious clues that can be obtained from the spacial domain, the frequency representation of perturbations show significant and systematic changes which can be exploited to detect attacks.

2018), Cleverhans (Papernot et al., 2016) and AdverTorch (Ding et al., 2019) were already available but did not include all SOTA methods in one evaluation.

The currently dominating adversarial attack method is *AutoAttack* (Croce & Hein, 2020a) which is an ensemble of 4 attacks: two variations of the PGD (Madry et al., 2018) attack with cross-entropy loss (APGD-CE) and difference of logits ratio loss (APGD-t), the targeted version of the FAB attack (Croce & Hein, 2020c), and the black-box Squares attack (Andriushchenko et al., 2020). The *AutoAttack* benchmark provides several modes. The "standard" mode executes the 4 attack methods consecutively. Only if one attack fails, the failed samples are handed over to the next attack method. The "individual" mode provides results for each attack on all input samples.

**Adversarial Training.** Adversarial Training (AT) can be backtracked to (Goodfellow et al., 2015), in which models were hardened by generating adversarial examples and adding them into training data. An adversarial example is a subtly modified image causing a machine learning model to misclassify it. The achieved robustness by AT depends on the strength of the adversarial examples used. E.g., training on Goodfellow's FGSM, which is fast and non-iterative, only provides robustness against non-iterative attacks, but e.g. not against PGD (Madry et al., 2018) attacks. Conse-

quently, (Tramèr et al., 2020) propose training on multi-step PGD adversaries, achieving SOTA robustness levels against $\ell_\infty$ attacks on MNIST and CIFAR10 datasets. Unfortunately, the high computational complexity of adversarial training makes it impractical for large-scale problems such as ImageNet.

**Adversarial Detection.** Many recent works have focused on adversarial detection, trying to distinguish adversarial from natural images. The authors in (Hendrycks & Gimpel, 2017) showed that adversarial examples have higher weights for larger principal components of the images' decomposition and use this finding to train a detector. Both (Li & Li, 2017) and (Bhagoji et al., 2017) leverage Principal Component Analysis (PCA) as well. Based on the responses of the neural networks' final layer (Feinman et al., 2017) define two metrics, the kernel density estimation and the Bayesian neural network uncertainty to identify adversarial perturbations. (Liu et al., 2019) proposed a method to detect adversarial examples by leveraging steganalysis and estimating the probability of modifications caused by adversarial attacks. (Grosse et al., 2017) used the statistical test of maximum mean discrepancy to detect adversarial samples. Using the correlation between images, based on influence functions and the k-nearest neighbors in the embedding space of the Deep Neural Network (DNN), (Cohen et al., 2020)

**Table 2** RobustBench: The top-5 entries of CIFAR10 leaderboard for $\ell_\infty$ in June 2021.

| Rank | Method | Standard Accuracy | Robust Accuracy | Extra data | Architecture | Date |
|------|--------|-------------------|-----------------|------------|--------------|------|
| 1 | Fixing Data Augmentation to Improve Adversarial Robustness | 92.23% | 66.56% | ✓ | WideResNet-70-16 | Mar 2021 |
| 2 | Uncovering the Limits of Adversarial Training against Norm-Bounded Adversarial Examples | 91.10% | 65.87% | ✓ | WideResNet-70-16 | Oct 2020 |
| 3 | Fixing Data Augmentation to Improve Adversarial Robustness | 88.50% | 64.58% | ✗ | WideResNet-106-16 | Mar 2021 |
| 4 | Fixing Data Augmentation to Improve Adversarial Robustness | 88.54% | 64.20% | ✗ | WideResNet-70-16 | Mar 2021 |
| 5 | Uncovering the Limits of Adversarial Training against Norm-Bounded Adversarial Examples | 89.48% | 62.76% | ✓ | WideResNet-28-10 | Oct 2020 |

proposed an adversarial detector. Besides the statistical analysis of the input images, adding a second neural network to decide whether an image is an adversarial example is another possibility. (Metzen et al., 2017) proposed a model that is trained on outputs of multiple intermediate layers. Two strong and popular detectors are the Local Intrinsic Dimensionality (LID) (Ma et al., 2018) and the Mahalanobis Distance (M-D) (Lee et al., 2018) detectors. Ma *et al.* used the LID as a characteristic of adversarial subspaces and identified attacks using this measure. Lee *et al.* computed the empirical mean and covariance for each training sample and then calculated the M-D distance between a test sample and its nearest class-conditional Gaussians.

**Fourier Analysis of Adversarial Attacks.** (Tsuzuku & Sato, 2019) showed that CNN are sensitive in the direction of Fourier basis functions, and proposed a Fourier-based attack method. Investigating trade-offs between Gaussian data augmentation and adversarial training (Yin et al., 2019) take a Fourier perspective and observe that adversarial examples are not only a high-frequency phenomenon. In (Ma et al., 2020) it is assumed that internal responses of DNN follow the generalized Gaussian distribution, both for benign and adversarial examples, but with different parameters. They extract the feature maps at each layer in the classification network and calculate the Benford-Fourier coefficients for all of these representations. This approach is similar to our white-box detector, but as our experiments show, it is more than sufficient to use our simplified features built on a standard 2D Discrete Fourier Transformation (DFT).

## 2. Methods: DFT based Detection

Our proposed detection method is based on the frequency-domain features originally introduced in (Harder et al., 2021), which we revise in the next subsections. In contrast to (Harder et al., 2021), we explicitly propose two types of detectors. First, a White-Box detector which has access to the feature maps of the target network, allowing it to observe the network response to input images, and second, a more general Black-Box detector which has no knowledge about the target network. We found that the Fourier power spectrum provides sufficient information to detect perturbations in both cases. Hence, we neglect the phase-based features which are also suggested in (Harder et al., 2021).

**Fourier Analysis.** The Fourier transformation decomposes a function into its constituent frequencies. A signal sampled at equidistant points is thereby known as discrete Fourier transform. The discrete Fourier transform of a signal with length $N$ can be computed efficiently with the Fast Fourier Transformation (FFT) in $\mathcal{O}(N \log N)$ (Cooley & Tukey, 1965). For a discrete 2D signal, like color image channels or single CNN feature maps – $X \in [0,1]^{N \times N}$ – the 2D discrete Fourier transform is given as

$$\mathcal{F}(X)(l,k) = \sum_{n,m=0}^{N} e^{-2\pi i \frac{lm+kn}{N}} X(m,n), \qquad (1)$$

for $l, k = 0, \ldots N-1$, with complex valued Fourier coefficients $\mathcal{F}(X)(l,k)$. In the following, we will only utilize the magnitudes of Fourier coefficients

$$|\mathcal{F}(X)(l,k)| = \sqrt{\text{Re}(\mathcal{F}(X)(l,k))^2 + \text{Im}(\mathcal{F}(X)(l,k))^2} \qquad (2)$$

and show that this information is sufficient to detect adversarial attacks with high accuracy.

### 2.1. Black-Box Detection: Fourier Features of Input Images

Figure 1 gives a brief visualization of the analysis of the changes in successfully perturbed images from *AutoAttack*: While different attacks show distinct but randomly located change patterns in the spatial domain (which makes them hard to detect), adversarial samples show strong, well-localized signals in the frequency domain.
Hence, we extract and concatenate the 2D power spectrum of each color channel (see eq. (2) and the right column of fig. 1) as feature representations of input images and use simple classifiers like Random Forests and Logistic Regression to learn to detect perturbed input images.

### 2.2. White-Box Detection: Fourier Features of Feature-Maps

In the white-box case, we apply the same method as in the black-box approach, but extend the inputs to the feature map responses of the target network to test samples. Since this extension will drastically increase the feature space for larger target networks, we select only a subset of the available feature maps. Note that the optimal selection of feature maps depends on the topology of the target network.

See table 4 for details on our selection for CIFAR10.

## 2.3. Measuring Adversarial Detection

The *AutoAttack* benchmark (Croce & Hein, 2020a), like most of the literature regarding adversarial robustness, uses a "Robust Accuracy" measure to compare different methods (see table 2 for details). However, our approach does not fit this evaluation scheme, since we are aiming to reject adversarial test samples instead of hardening the networks. Therefore, we report two different indicators: The *Adversarial Succes Rate (ASR)* in eq. (3) is calculated as

$$\text{ASR} = \frac{\text{\# perturbed samples}}{\text{\# all samples}} \qquad (3)$$

the fraction of successfully perturbed test images and provides a baseline of *AutoAttack*'s ability to fool unprotected target networks.

We measure the performance of our defense by the *Adversarial Success Rate under Detection (ASRD)* in eq. (4). Here, we compute the ratio of successful attacks under defense

$$\text{ASRD} = \frac{\text{\# undetected perturbations}}{\text{\# all samples}} = \text{FNR} \cdot \text{ASR}, \quad (4)$$

where FNR is the false negative rate of the applied detection algorithm. ASR is only a scaling factor for the FNR. The lower the ASRD rate, the more pertubated examples are defeated.

## 3. Experiments

**CIFAR10.** We trained a WideResNet28-10 (Zagoruyko & Komodakis, 2017) on the CIFAR10 training set to a test-accuracy of 94% and applied *AutoAttack* on the test set. Then we extracted the spectral features and used a random subset of 1500 samples of this data for each attack method to train and evaluate our base classifiers (train:test split 80:20). Table 1 shows results using *AutoAttack* in "standard" mode for various $\varepsilon$ on $\ell_\infty$-perturbations. Table 3 shows the results for the "individual" mode for $\varepsilon = 8/255$ and $\ell_\infty$-perturbations as well as a comparison to other detection methods. Here we used 1000 samples from each dataset, CIFAR10 and ImageNet, for our evaluation.

**ImageNet.** For the benchmarks on ImageNet we used the pre-trained WideResNet51-2 (Zagoruyko & Komodakis, 2017) from the PyTorch library. As test set, we apply the official validation set from ImageNet. The accuracy of this pre-trained model is about 78%. As for CIFAR10, *AutoAttack* also shows strong adversarial performance on ImageNet. Table 1 shows very high ASR even for low $\varepsilon$ values. Note that $\varepsilon < 0.5$ would not represent realistic attack scenarios since saving the perturbed images would round the adversarial changes to the next of 256 available

**Table 3** $F1$-score comparison of detection methods. The attacks from *AutoAttack* individual mode are applied using the CIFAR10 test set on a WideResNet28-10 with $\varepsilon = 8/255$. Logistic regression has been used as base classifier.

| Dataset | Detector | Attack | | | |
|---|---|---|---|---|---|
| | | apgd-ce | apgd-t | fab-t | square |
| CIFAR10 | on WideResNet28-10 | | | | |
| | Black-Box | 94 | 91 | **60** | 64 |
| | White-Box | **96** | **93** | 55 | **75** |
| | LID | 53 | 47 | 49 | 48 |
| | M-D | 49 | 49 | 46 | 48 |
| ImageNet | on WideResNet51-2 | | | | |
| | Black-Box | 82 | 77 | 60 | 78 |
| | White-Box | **97** | **96** | 52 | **92** |
| | LID | 61 | 60 | 45 | 59 |
| | M-D | 60 | 62 | **66** | 57 |

**Table 4** Comparison of individual features from different layers via $F1$-score. Individual AutoAttack. Base classifier is logistic regression. $\varepsilon = 8/255$.

| CIFAR10 | on WideResNet28-10 | | | | |
|---|---|---|---|---|---|
| Layers | Dim. | Attack | | | |
| | | apgd-ce | apgd-t | fab-t | square |
| conv2 0 WB | 32768 | 93 | 88 | **58** | 75 |
| conv2 1 WB | 327680 | 94 | 89 | 55 | 77 |
| conv2 2 WB | 327680 | 93 | **93** | 56 | 75 |
| conv2 3 WB | 327680 | 93 | 92 | 57 | **85** |
| conv3 0 WB | 327680 | **96** | 89 | 56 | 78 |
| conv3 1 WB | 163840 | 77 | 64 | 45 | 67 |
| conv3 2 WB | 163840 | 72 | 61 | 47 | 64 |
| conv3 3 WB | 163840 | 75 | 65 | 50 | 69 |
| conv4 0 WB | 163840 | 78 | 65 | 45 | 70 |
| conv4 1 WB | 81920 | 69 | 54 | 47 | 58 |
| conv4 2 WB | 81920 | 70 | 56 | 48 | 55 |
| conv4 3 WB | 81920 | 68 | 56 | 46 | 61 |
| relu | 81920 | 68 | 56 | 50 | 60 |

bins in commonly used 8-bit per channel image encodings. Table 3 shows the results for individual attacks.

## 4. Discussion and Conclusion

In this paper, we are able to show a first proof of concept that simple frequency features can be used to detect current SOTA attacks with a very high accuracy on the standard CIFAR10 benchmark and on a the more complex ImageNet dataset. Especially the black-box approach could provide a practical counter-measure for the defense of real-world applications. However, there are still many open questions: I) How well will the detectors generalize to other datasets, network architectures, and new attacks? II) Why is *AutoAttack* so successful for very small $\varepsilon$ in ImageNet? III) Can the detection be combined with Adversarial Training methods like the ones shown in table 2? In light of these open questions, we expect our approach can build a solid basis for future research.

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

# Appendix

**Table 5** Extension of table 1, showing full results for Logistic Regression (LR) and Random Forest (RF). RF outperforms LR in almost all cases except the White-Box approach on ImageNet. Due to the large feature-space, RF appear to be overfitting in this case.

| $\varepsilon$ | ASR | Black-Box | | | | | | White-Box | | | | | |
|---|---|---|---|---|---|---|---|---|---|---|---|---|---|
| | | F1 | | FNR | | ASRD | | F1 | | FNR | | ASRD | |
| | | LR | RF | LR | RF | LR | RF | LR | RF | LR | RF | LR | RF |
| CIFAR10 on WideResNet28-10 | | | | | | | | | | | | | |
| 8/255 | 100 | 98.0 | **98.2** | 00.0 | **00.0** | 00.0 | **00.0** | 97.2 | **99.0** | 02.7 | **00.0** | 02.7 | **00.0** |
| 4/255 | 100 | **94.8** | 93.8 | 15.0 | **00.3** | 15.0 | **00.3** | 83.7 | **96.4** | 18.0 | **05.0** | 18.0 | **05.0** |
| 2/255 | 93.1 | **86.3** | 85.0 | 28.7 | **05.3** | 26.7 | **04.9** | 62.9 | **85.8** | 35.0 | **05.0** | 32.6 | **04.6** |
| 1/255 | 49.6 | **74.0** | 70.8 | 41.3 | **22.7** | 20.5 | **11.3** | 54.8 | **62.5** | 46.0 | **37.3** | 22.8 | **18.5** |
| 0.5/255 | 12.3 | 54.4 | **54.8** | 48.3 | **46.7** | 06.0 | **05.8** | 51.7 | **61.6** | **50.0** | 52.0 | **06.0** | 06.4 |
| ImageNet on WideResNet51-2 | | | | | | | | | | | | | |
| 8/255 | 100 | 81.9 | **87.1** | 16.3 | **00.7** | 16.3 | **00.7** | **96.7** | 90.4 | **01.3** | 03.0 | **01.3** | 03.0 |
| 4/255 | 100 | 65.0 | **77.4** | 36.0 | **08.7** | 36.0 | **08.7** | **94.1** | 82.3 | **02.7** | 07.7 | **02.7** | 07.7 |
| 2/255 | 100 | 55.9 | **60.2** | 44.3 | **28.0** | 44.3 | **28.0** | **82.3** | 73.5 | **16.7** | 20.7 | **16.7** | 20.7 |
| 1/255 | 99.9 | 50.8 | **53.4** | 50.7 | **43.7** | 50.6 | **43.6** | **67.9** | 59.1 | **30.7** | 40.7 | **30.6** | 40.6 |
| 0.5/255 | 96.9 | **54.7** | 54.4 | **40.7** | 42.0 | **39.4** | 40.7 | **59.0** | 50.5 | **41.0** | 50.0 | **38.1** | 48.5 |