# OpenReview forum: "Detecting AutoAttack Perturbations in the Frequency Domain"
_ICML.cc/2021/Workshop/AML — ICML 2021 Workshop AML Poster_

### Official Review · Reviewer_k711 · 2021-06-19
**The paper proposed a simple adversarial detection method based on frequency domain analysis. It looks simple, but is effective and reasonable.**

**Rating:** Accept
**Confidence:** 4

**Review:**

The paper proposed a simple adversarial detection method based on frequency domain analysis. The proposed method consists of two manners. In the black-box manner, the paper extracts and concatenates the 2D power spectrum of each color channel as feature representations of input images, and uses the feature for the classification. And in the white-box manner, it extend the inputs to the feature map responses of the target network to test samples. The experimental results show the effectiveness of the proposed method.

pros:
1. The writing is good and is easy to follow.
2. Although the proposed method in the paper is simple, it is effective and reasonable.
3. The experimental results are sufficient.

cons:
1. 1000 samples for CIFAR10 is not very sufficient. It‘s better to increase the data size.
2. Typo in Line 166: AutoAttackindividual ->  AutoAttack individual

---

### Decision · Program_Chairs · 2021-06-21

**Decision:**

Accept (Poster)

**Comment:**

The paper proposed a simple adversarial detection method based on frequency domain analysis. The authors can further address the reviewer's comment.